# Generative-Adversarial-Network-Based Image Reconstruction for the Capacitively Coupled Electrical Impedance Tomography of Stroke

**DOI:** 10.3390/life14030419

**Published:** 2024-03-21

**Authors:** Mikhail Ivanenko, Damian Wanta, Waldemar T. Smolik, Przemysław Wróblewski, Mateusz Midura

**Affiliations:** Faculty of Electronics and Information Technology, Warsaw University of Technology, 00-665 Warsaw, Poland; mikhail.ivanenko.dokt@pw.edu.pl (M.I.); waldemar.smolik@pw.edu.pl (W.T.S.); przemyslaw.wroblewski@pw.edu.pl (P.W.); mateusz.midura.dokt@pw.edu.pl (M.M.)

**Keywords:** capacitively coupled electrical impedance tomography, image recognition, inverse problem, machine learning, neural networks, cGAN, brain imaging, stroke detection

## Abstract

This study investigated the potential of machine-learning-based stroke image reconstruction in capacitively coupled electrical impedance tomography. The quality of brain images reconstructed using the adversarial neural network (cGAN) was examined. The big data required for supervised network training were generated using a two-dimensional numerical simulation. The phantom of an axial cross-section of the head without and with impact lesions was an average of a three-centimeter-thick layer corresponding to the height of the sensing electrodes. Stroke was modeled using regions with characteristic electrical parameters for tissues with reduced perfusion. The head phantom included skin, skull bone, white matter, gray matter, and cerebrospinal fluid. The coupling capacitance was taken into account in the 16-electrode capacitive sensor model. A dedicated ECTsim toolkit for Matlab was used to solve the forward problem and simulate measurements. A conditional generative adversarial network (cGAN) was trained using a numerically generated dataset containing samples corresponding to healthy patients and patients affected by either hemorrhagic or ischemic stroke. The validation showed that the quality of images obtained using supervised learning and cGAN was promising. It is possible to visually distinguish when the image corresponds to the patient affected by stroke, and changes caused by hemorrhagic stroke are the most visible. The continuation of work towards image reconstruction for measurements of physical phantoms is justified.

## 1. Introduction

Electrical impedance tomography with capacitively coupled electrodes (CCEIT) [1,2,3] has greater potential in brain imaging than classic electrical impedance tomography (EIT) [4,5,6,7] due to problems with the impedance of the electrode–skin contact [8,9]. In EIT, four-electrode measurement limits the influence of electrode impedance on the measurement, but compensation is insufficient if an electrode impedance mismatch occurs [10,11,12]. In CCEIT, a two-electrode measurement is performed in the form of electrical capacitance tomography (ECT) [13,14,15]. The contactless electrodes are insulated from the skin by a dielectric and do not require gluing to the skin with gel. This is of great importance in medical practice because it facilitates the performance of tests and increases patient comfort. However, coupling capacitance may affect the measurement. The coupling capacitance value must be high to make the contact susceptance significant compared to the measured conductivity. The changes in the coupling capacitance value during the measurement also disturb the results, but the monitoring of coupling capacitance during measurement is theoretically possible [16].

The application of electrical impedance tomography in brain imaging, especially in stroke detection [17,18], attracts a lot of interest due to its relatively low cost, absence of side effects, and the possibility of developing wearable solutions [19]. Poor blood flow to the brain tissues during a stroke may be caused by a lack of blood flow (ischemic stroke) or bleeding (hemorrhagic stroke) [20,21,22]. It has been shown that the conductivity of tissues in stroke-affected areas is significantly different from that of healthy tissues and differs in both stroke types [18,23,24]. However, it should be remembered that a significant barrier in brain impedance measurements is the very low conductivity of the skull bones [25]. According to the guidelines, stroke differentiation should be based on X-ray computed tomography (CT) or magnetic resonance imaging (MRI) [25]. However, even nowadays, the availability of such equipment could be limited, especially in underdeveloped regions. Despite its low spatial resolution, we believe that EIT can find its place in brain imaging when CT and MRI modalities are unavailable.

The factors limiting the quality of EIT images are a low signal level with high signal dynamics for adjacent and opposite electrodes, poor sensitivity to changes in electrical parameters in the center of the examined object, a small number of electrodes, poor spatial sampling, as well as an ill-posed and numerically ill-conditioned nonlinear inverse problem [4]. Image reconstruction in EIT is severely ill-posed due to the relatively small number of measurements in comparison to the number of reconstructed pixels. However, it is possible to find a satisfactory solution using a wide variety of algorithms, from simple algebraic ones, like linear back projection (LBP), to modern machine-learning-based methods [26,27,28,29]. The application of machine learning (ML) and neural networks for EIT image reconstruction and classification is widely explored [30,31,32,33,34]. The idea of synthetically generating training samples and using them to train the neural network has been employed for 30 years [35]. The interest in machine-learning-based approaches arose because algebraic methods either provide poor-quality reconstruction or require heavy computational power due to extremely poor numerical conditioning, which impedes inverse problem regularization [36,37]. It is possible to use various ML-based algorithms in EIT image reconstruction, such as artificial neural networks, Random Forests, K-Nearest Neighbors, Elastic Net, Ada Boost, and Gradient Boosting [38]. In our work, we focused on ANN approaches as they show very promising results in the reconstruction of medical EIT images, for example, the fully connected network [39] and hybrid convolutional network [40] for brain EIT or the conditional generative adversarial network (cGAN) for thoracal EIT [33]. Also, ANN-based classifiers can support stroke differentiation-related decisions [41,42].

The cGAN approach is widely used in EIT image reconstruction. We have already successfully confirmed the possibility of using cGAN in thoracal CCEIT [15]. However, according to our knowledge, there have been no attempts to explore its feasibility for human brain CCEIT image reconstruction. Inspired by the study for brain EIT focused on intracerebral hemorrhage imaging [40], we propose a new approach for more precise brain modeling and apply a cGAN-based algorithm for image reconstruction in human brain CCEIT.

This paper presents the experimental verification of adversarial network image reconstruction in the capacitively coupled electrical impedance tomography of brain stroke. The experiments were conducted numerically using a computer simulation of tomographic measurements. To conduct the simulation, we developed a numerical phantom of the human head inspired by the famous Shepp–Logan phantom used in X-ray computed tomography simulations [43]. This study aimed to verify whether image reconstruction based on an artificial neural network (cGAN) allows the reconstruction of acceptable brain images and the detection of stroke regions from measurements made using the CCEIT technique.

## 2. Materials and Methods

This work’s primary focus is exploring conditional generative adversarial networks’ potential for human brain CCEIT image reconstruction. As CCEIT is a new emerging technique that has not yet been introduced into clinical practice, numerical simulation is the only possible way to acquire training data. In this section, we describe a method for the simulation of tomographic measurements and introduce the neural network architecture. Dataset generation and the network training procedure are described in subsequent sections.

### 2.1. Numerical Simulation of Tomographic Measurement

When using supervised machine learning, it is necessary to acquire a large dataset containing samples of input and expected output data. In the brain CCEIT, input data are capacitance measurements of a human head, and output data are a corresponding spatial distribution of electrical properties in the brain. Collecting a set of several tens of thousands of patient studies is unrealistic. No database of such reference data exists. However, it is possible to generate the required data using numerical simulation. To solve a forward problem in EIT, we needed to develop a numerical phantom of the human brain. This could help us define the distribution of expected electrical properties inside the brain. Having the phantom defined, we could vary the phantom elements’ position and size, generating any number of samples describing conductivity and electrical permittivity.

To generate expected capacitance measurements by the given distribution of electrical properties, we used the ECTSim v1.1 toolbox running in the Matlab environment [44]. This toolbox calculates complex capacitance as an output and expects complex electrical permittivity as an input:(1)$$\varepsilon =\varepsilon^{\prime }-j\frac{\sigma }{\omega }$$
where ε′ is permittivity, σ is conductivity, and ω is the angular frequency. The measurements are calculated using the Gauss law equation:(2)$$C_{m\left(i,j\right)}=\frac{1}{V_{i}-V_{j,\;j\neq i}}\underset{\partial \Omega_{j}}{∯}\varepsilon \left(r\right)E\left(r\right)ds$$
applying the finite volume method [45]. $\varepsilon \left(r\right)$ is complex permittivity at position r in the examined volume, $E\left(r\right)$ is the electric field, $\partial \Omega_{j}$ is a surface surrounding the measuring electrode j, ds is a normal vector to a small element of the surface, $V_{i}$ is a potential at the i-th electrode and $C_{m\left(i,j\right)}$ is a complex capacitance measured for a pair of electrodes $\left(i,j\right)$.

Our experiment numerically simulated the actual CCEIT device used in our laboratory. The bioimpedance measurement method used by our system is described in detail in [16]. We simulated a two-wire measurement method with pulse excitation. Figure 1a shows the electronic equivalent circuit of the simulated measurement system. The bioimpedance of the head $Z_{x}$ is represented as a parallel connection of capacitance $C_{x}$ and resistance $R_{x}$. The serially connected coupling capacitance $C_{c}$ results from the dielectric layer insulating the electrodes. Since, in the measurement, the smaller capacitance dominates in a series of capacitance connections, the contact capacitance must be large compared to the measured impedance. This means that, in the case of bioelectrical measurements of a medium with low resistance, the electrode’s insulation should be as thin as possible and made of a material with high electrical permittivity. The air gap between the insulation and the surface of the head should be minimized.

To simulate the bioimpedance measurement of the human head, we modeled a 16-electrode measuring belt with surface electrodes covered with an insulation layer (Figure 1b). Belt dimensions correspond to the average size of a human head [46]. In the simulation, we considered the measurement range of our device, the noise level, and the influence of the coupling capacitance on the deterioration of the measurement quality.

### 2.2. Artificial Neural Networks in CCEIT Image Reconstruction

A forward problem in CCEIT relies on representing capacitance measurements and the electrical permittivity distribution as vectors of corresponding sizes and then finding a transformation operator representing dependence between those vectors:(3)c=f(ε)

Such an operator is nonlinear, but it is possible to use its linear approximation in the simplified case. In this case, we can represent the forward problem by multiplying the electric permittivity distribution vector by a sensitivity matrix:(4)∆c=∆εS
where $∆\varepsilon \in R^{N}$ is a vector representing the electrical permittivity change in each of the *N* pixels, $∆c\in R^{M}$ is a vector representing capacitance change due to permittivity change at *M* measurements, and $S\in R^{MxN}$ is a sensitivity matrix, which is a Jacobian, showing how the change in electrical permittivity in each pixel influences each capacitance measurement.

Solving the inverse problem implies, in this case, finding an inverse of the matrix *S*. However, since matrix *S* is not a square matrix and, therefore, is not strictly invertible, it is necessary to use approximate inverse methods as follows: the Moore–Penrose pseudoinversion or matrix transposition could replace the matrix inverse operator, as in the linear back projection method. Nevertheless, as a rough linear approximation, this approach only shows satisfactory results in relatively simple object imaging. In more complex cases, for example, human body imaging, the nonlinear nature of the operator f (3) becomes essential.

Since artificial neural networks with the unbounded activation function have the property of being a universal approximator [47], it is possible to use ANN to represent the nonlinear operator f and use such a network to solve the inverse problem in CCEIT. There are two main challenges while training the network, as follows: choosing network architecture and obtaining the training dataset. The minimal neural network for EIT image reconstruction consists of linear layers since artificial neural networks with the unbounded activation function have the property of being the universal approximator [47], and it is possible to use ANN to represent the nonlinear operator f and use such a network to solve the inverse problem in CCEIT. Two main challenges while training the network are choosing the network architecture and obtaining the training dataset. The minimal neural network for EIT image reconstruction consists of linear layers (5) and ReLU (rectified linear unit) or its generalized version leaky ReLU activation function:(5)$$y(x)=xW^{T}+b$$
(6)$$LeakyReLU(x)=\left\{\begin{array}{c} \alpha x,\;\;x<0\\ x,\;\;x\geq 0 \end{array}\right.$$
where W—layer learnable weights and b—layer learnable bias. The activation function adds necessary nonlinearity and allows ANN to approximate nonlinear functions.

More sophisticated networks, such as U-Net-based networks [48], additionally contain convolution and deconvolution layers. The basic 2D convolution layer is shown in Figure 2. The deconvolution layer, or transposed convolution layer, is a reciprocal operation to convolution and is usually implemented using the data gradient operation. Convolution and deconvolution layers can be of an arbitrary number of dimensions, but the 2D version is the most useful for image processing and, therefore, for image reconstruction.

To increase network stability and make training faster, it is possible to add batch normalization layers [49]:(7)$$y(x)=\frac{x-Mean[x]}{\sqrt{Var\left[x\right]+\epsilon }}W+b$$
where W—layer learnable weights, b—layer learnable bias, Mean[x]—moving average, Var[x]—moving variance, and ϵ—constant for numerical stability.

In addition to the ANN architecture itself, the training procedure is also essential. The conditional generative adversarial network (cGAN) approach is very useful in image reconstruction. It is based on the idea of adding the simple classifier network, named the discriminator, to the main network, named the generator. After that, it is possible to train both networks by exposing ground truth images and reconstructed images to the discriminator and constructing the loss function so that an output value of 1 is expected when the true image is presented. When the reconstructed image is given, an output value of 0 is expected from the discriminator. The network trains successfully when the loss function tends to 0.5, which means that the discriminator cannot distinguish between the true and reconstructed images produced by the main generator network.

### 2.3. Neural Network Architecture

Our previous work [15] showed that modified cGAN-based neural network architecture inspired by the Pix2Pix approach [50] is very promising for electrical tomography image reconstruction. Previously, this network was trained on the imaginary component of normalized measurement data obtained by simulating measurements for an empty sensor and the sensor filled with high-conductivity material. In this work, we trained the real-valued network using the raw measurements’ real and imaginary components, with the input matrix size of the algorithm as 2 × 240 and the output matrix size as 64 × 64.

One of the advantages of the chosen network architecture is that the number of weights of the network layers depends almost exclusively on the output image size. Changing the size of the input data vector requires only changes in the number of weights of the very first linear layer of the generator (Figure 3) and discriminator (Figure 4).

However, using a 16-electrode sensor instead of a 32-electrode sensor dramatically shrinks the size of the input vector from 992 to 240. Even using both real and imaginary components, what makes the size of the input vector 1 × 480 requires network training procedure tuning. Specifically, the most important step is to prevent discriminator loss from collapsing to zero because when this happens, the discriminator becomes too strong and can easily distinguish the reconstructed image from the real one with a near 100% probability.

## 3. Numerical Phantom

Supervised neural network training requires having a large dataset representing the dependence between the network input and expected output. In the case of CCEIT image reconstruction, the network input is a set of capacitance measurements, and the output is a corresponding conductivity distribution. As mentioned, the CCEIT technique is still new and has not yet been introduced into clinical practice compared to EIT. Therefore, obtaining such a dataset containing human brain CCEIT measurements is only possible using numerical simulation at the current level of CCEIT development.

It is possible to create a simplified geometrical model representing the human head and use such a numerical phantom to generate a training dataset of an arbitrary number of samples. Our study shows the method of creating a training dataset with an arbitrary number of samples representing a possible human head based on selected real-life MRI examinations (Figure 5). We can infer brain model geometric constraints using a few examinations and generate a large dataset containing samples within defined limits.

### 3.1. Model of the Human Head

A numerical human head phantom was developed based on the IXI dataset [51], containing about 600 T1 MRI examinations of healthy subjects. After aligning all dataset samples and segmenting them into the skull, grey matter, white matter, and cerebrospinal fluid (CSF), the three-centimeter-thickness cross-section volume was cut from segmented 3D images (Figure 6). Then, corresponding conductivity and electrical permittivity values, according to Table 1, were assigned to voxels belonging to particular tissues. The mean value was calculated throughout cross-sections to obtain the mean value image of the volume. In the numerical model, we assumed the dielectric properties for a frequency of 64 MHz, which allowed the comparison of the results with 1.5 T electrical properties tomography [52]. Recently proposed electrical properties tomography (EPT) is based on magnetic resonance image processing and allows the precise reconstruction of electrical properties distribution in the brain. Several EPT brain studies showed visible changes in brain electrical properties in the regions affected by stroke [53,54].

Four regions were defined for the modeling as follows: the skull and skin together, two brain hemispheres, and CSF. The regions were modeled using ellipses and positioned according to the mean value images of electrical properties, as shown in Figure 7. Inter-individual variability was considered by measuring the position and size of the ellipses throughout the 50 selected images from the dataset and then determining the possible range of values. Mean conductivity and electrical permittivity values for the regions were calculated for each of the 50 samples. Subsequently, after removing outliers by filtering out values with more than triple the standard deviation, the mean and standard deviation were calculated for each region (Table 2) for training the dataset generation. The distribution of the values throughout all 50 samples is shown in Figure 8.

We defined the elements of the 2D numerical model of the human head axial cross-section using geometrical primitives such as ellipses with variable positions, sizes, and tilt. The 2D model is the result of averaging MRI cross-sections in the three-centimeter range corresponding to the height of the measurement electrodes in the CCEIT sensor. When developing our measuring belt model, we assumed a head circumference of 575 mm, which corresponds to the size of an average man’s head [46]. In numerical simulations, the constant head geometry was represented by the ellipses with semiaxes measuring 77 mm and 105 mm, which gave the required circumference. The fixed-size ellipse was positioned at point (0,0) and was collinear to the image’s vertical and horizontal axes. Two inner ellipses defined by tissue thickness were added to represent the skin and skull bone. Electrical parameters in regions corresponding to the skin and skull bone were assigned according to Table 1. Each brain hemisphere was modeled by two ellipses with variable center positions, axes sizes, and tilt (Figure 9a,b). Both hemispheres were limited by the additional ellipse (‘Inner’ object in Table 3), preventing the intersection of the brain region with the elements representing the skin and skull. The brain was divided into two hemispheres by the vertical line shifted in the X direction at −3.0 ± 6.0 mm and in the Y direction at −8.5 ± 10.5 mm and tilted relative to the Y axis at 1.0 ± 10.0 deg. Cerebrospinal fluid was modeled by a composition of the five ellipses (Figure 9c), where the central larger one had variable position, size, and tilt. The angular position was chosen for each of the other four ellipses of varying size and tilt. Then, the ellipse’s center was positioned at the variable distance on the outward-pointing normal vector to the first ellipse at the point corresponding to the chosen angular position. A complete list of the model elements and their parameters is shown in Table 3.

Two approaches were used to generate dataset samples to increase variability from one side and better represent reality from the other. Both approaches are based on manually fitting the model to 3 cm thick MRI slices. The first approach relies on finding constraints on elements’ size and position based on manually marked models. A complete list of the geometric constraints used in the model is shown in Table 3. Additionally, it was necessary to apply more constraints to prevent the separation of the model elements representing the same tissue.

The ellipse’s circumference and axes ratio were limited, as shown in Table 4. The brain area was limited to 23.0 ± 6.0% of the total image area, the CSF area was limited to 23.0 ± 6.0%, and the area of the rest of the head was limited to 13.5 ± 5.5%. The ratio between the left and right hemispheres area was constrained to 98.5 ± 7.5%.

The second approach relies on randomly selecting one of the models and modifying all model parameters, adding random numbers from the normal distribution with a relatively small standard deviation.

Conductivity and permittivity values are selected randomly according to Gaussian distribution, with the mean and standard deviation shown in Table 2. An additional condition is applied, ensuring that values for the CSF region are greater than the brain region and that values for the brain region are greater than for the skin and skull region.

### 3.2. Training Dataset with Diseases

Stroke was modeled using a circular region with altered electrical properties added to the left or right brain hemisphere. In the case of hemorrhagic stroke, conductivity and permittivity values were equal to 1230 $\left[\frac{mS}{m}\right]$ and 76, correspondingly [55]. The conductivity value corresponded to the conductivity of blood. While modeling ischemic stroke, the permittivity value was assumed to be the same as the surrounding brain tissue, and conductivity was randomly selected as 65 ± 5% of the conductivity of the surrounding brain tissue. We expected that, with such parameters, it would be much more challenging to detect ischemic than hemorrhagic stroke.

The position of the stroke region was selected randomly using the following procedure. Six lines were defined to model stroke-affected cases, representing the main brain arteries according to known supply areas of the cerebral arteries [56]. Lines starting from the point in the middle of the main axis’s upper part were tilted at the following angles: 12, 80, 135, −12, −80, −135 degrees relative to the vertical axis. Ischemic stroke was modeled by selecting one of the arteries and drawing a circular region on the line representing this artery but with the center outside the brain. The intersection of this region and the cerebral hemisphere was treated as the stroke region. Hemorrhagic stroke was modeled by selecting a point on the line representing the blood vessel inside the region representing the brain. The selected point was considered to be the center of the circle representing the stroke. The stroke size was selected randomly from 15 mm up to the whole distance between the CSF region and the edge of the brain region. The CSF region was subtracted from the stroke region, and the result was intersected with a corresponding hemisphere region to ensure it was located exclusively inside the brain region (Figure 10).

When generating dataset samples, we simulated 100,000 cases using the constraint-based model and 100,000 cases using the second approach based on introducing random disturbances into models based on actual samples. The patient condition was chosen randomly, resulting in 66,758 samples representing healthy individuals and 66,468 and 66,774 samples representing patients affected by hemorrhagic and ischemic stroke, respectively. To assess the reconstruction quality more reliably, we also generated a validation dataset of 50,000 simulated samples using the abovementioned approaches.

## 4. Network Training

After training and validating the dataset generation, we conducted neural network training according to the procedure shown in Figure 11. During training, the training part of the dataset was randomly divided into training and testing parts in the ratio of 75:25. The testing part of the dataset was used to control training and fine-tune network metaparameters, such as the discriminator and generator learning rates.

One of the Pix2Pix approach’s techniques is setting up different starting learning rates separately for the generator and discriminator. When the generator and discriminator’s starting learning rates were set to 10^−3^ and 10^−8^ correspondingly, the network became satisfactory and stable in cases where noise was added to the training data and when it was not (Figure 12).

Gaussian noise was added to the input data, providing a peak signal-to-noise ratio of 30 dB to increase the network’s robustness [57]. After tuning the learning rates described above, the network was trained for 100 epochs using a single NVIDIA Tesla P100 GPU (Nvidia Corporation, Santa Clara, CA, USA). The training time was about 5.5 h. During training, the mean relative standard error (8) was calculated on the testing part of the training dataset (Figure 13).
(8)$$IE=\frac{\left\|\hat{y}-y\right\|}{\left\|y\right\|}$$

To evaluate the network training quality, we adopted several simple pixel-to-pixel metrics such as the L2 norm (root mean square error, RMSE), the 2D correlation coefficient, peak signal-noise ratio (PSNR), and structural similarity index (SSIM) [58,59] defined as follows:(9)$$RMSE\left(\hat{y},y\right)=\sqrt{\frac{1}{N}\sum_{i=1}^{N}{\left|y_{i}-{\hat{y}}_{i}\right|}^{2}}\;$$
(10)$$CC\left(\hat{y},y\right)=\frac{\sum_{i=1}^{N}\left({\hat{y}}_{i}-\bar{\hat{y}}\right)\left(y_{i}-\bar{y}\right)}{\sqrt{\sum_{i=1}^{N}{\left({\hat{y}}_{i}-\bar{\hat{y}}\right)}^{2}\sum_{i=1}^{N}{\left(y_{i}-\bar{y}\right)}^{2}}}$$
(11)$$PSNR\left(\hat{y},y\right)=10\log_{10}N\frac{\max\hat{y}}{\sum_{i=1}^{N}{\left|y_{i}-{\hat{y}}_{i}\right|}^{2}}$$
(12)$$SSIM\left(\hat{y},y\right)=\frac{\left(2\mu_{y}\mu_{\hat{y}}+c_{1}\right)\left(2\sigma_{y\hat{y}}+c_{2}\right)}{\left(\mu_{y}+\mu_{\hat{y}}+c_{1}\right)\left(\sigma_{y}^{2}+\sigma_{\hat{y}}^{2}+c_{2}\right)}\;\;$$
where

N—the number of pixels,

$y_{i}$—the expected conductivity value at pixel i,

${\hat{y}}_{i}$—the reconstructed conductivity value at pixel i,

$\mu_{y}$—the mean value of vector y,

$\mu_{\hat{y}}$—the mean value of vector $\hat{y}$,

$\sigma_{y}^{2}$—the variance of vector y,

$\sigma_{\hat{y}}^{2}$—the variance of vector $\hat{y}$,

$\sigma_{y\hat{y}}$—the covariance of vectors y and $\hat{y}$,

$c_{1}$ and $c_{2}$—stabilization variables to prevent division by zero on weak denominators.

After training, we calculated an average of adopted metrics throughout the validation dataset to estimate reconstruction quality. We calculated a metric dispersion alongside a mean value because averaging did not consider outlying cases.

## 5. Results

As a result, an ANN-based model was trained, and reconstruction quality was assessed using the pixel-to-pixel metrics mentioned above, such as RMSE, PSNT, SSIM, and the 2D correlation coefficient. These metrics were calculated for each pair of the ground truth and reconstructed normalized images for different levels of noise applied to the simulated measurements (Table 5). The reconstruction examples for a healthy brain and brain affected by a hemorrhagic and ischemic stroke for the various noise levels are shown in Figure 14, Figure 15 and Figure 16. Additionally, histograms of image quality metrics distribution were created to reflect the reconstruction quality (Figure 14).

It is easy to note that image reconstruction for cases involving hemorrhagic stroke is much easier than for cases involving ischemic stroke. This was expected since ischemic stroke implies only subtle changes in the conductivity. However, in many samples from the test dataset, such changes are visible in the reconstructed images. Also, it is essential to emphasize that when the signal-to-noise level decreases, it becomes harder to reconstruct the image. But, having in mind that achievable SNR during actual measurements is about 30 dB, we can state that reconstruction quality at that noise level remains at a satisfactory level and, according to the metric distributions shown in Figure 17, it is not very different from the almost perfect signal (with an SNR of 60 dB)-based reconstruction quality.

## 6. Discussion

Electrical impedance tomography has already been introduced into clinical practice, and various image reconstruction methods have been developed. Such methods include LBP, Tikhonov regularization, Moore–Pensore pseudoinverse, the Levenberg–Marquardt iterative algorithm, and more sophisticated ones such as Calderon’s and the D-Bar methods. The main limitations of these methods are either poor reconstruction quality or the high computational power required to achieve satisfactory results. In the last decade, the evolution of rapid machine learning has contributed to the development of various artificial neural network-based approaches for EIT image reconstruction. They advanced from simple multilayer fully connected networks such as EIT-4LDNN [59] to complex hybrid convolutional networks [40]. The cGAN scheme, in the last few years, has become a widely adopted approach for training convolutional networks for EIT image reconstruction.

However, the new emerging capacitively coupled EIT technique, which is potentially more patient-friendly due to the lack of direct electrode–skin contact, requires deeper study before its introduction into clinical practice. We combined the most promising approaches, such as the cGAN procedure for ANN training and the U-Net-based generator network for simulating human brain CCEIT image reconstruction. We additionally modified the training procedure using the Pix2Pix approach initially developed to generate high-resolution images. To effectively train ANN, collecting a large dataset of conductivity distribution and corresponding capacitance measurements is necessary. However, it is impossible to collect real data measurements because it would require collecting thousands of examinations using techniques not yet adopted into clinical practice. Therefore, developing a numerical phantom that allows a synthetically generated large training dataset is necessary.

Our work focused on developing the human head numerical phantom for the CCEIT simulation. We proposed a method for determining phantom parameters using MRI head tomographic slices. Using the CCIET measurement numerical simulation, we prepared the training dataset with random samples, which allowed us to conduct convolutional neural network-supervised training that reconstructed synthetic cross-sectional brain images.

Applying the Pix2Pix approach to cGAN training allowed us to reconstruct 64 × 64 conductivity images from 16 electrode capacitance measurements comparable with EIT studies. However, our method has great potential in increasing output image resolution as it requires adding the additional convolutional block to the underlying U-Net generator.

Despite visible artifacts, the overall quality of the reconstruction appeared to be satisfactory enough to distinguish images representing the stroke-affected brain, especially in the case of hemorrhagic stroke. Even if the ability of the technique to discover ischemic stroke is limited, it is possible to use it not only to find the stroke but also to differentiate the stroke type. This might be crucial for further treatment-related decisions.

Comparing the quality measures obtained on the brain image dataset with the results obtained on the thorax dataset [15], it is essential to notice that brain reconstruction appears to be a more complicated task for the neural network. This was expected since the brain is surrounded by a skull with a much lower conductivity and electrical permittivity. Our numerical simulation proves that it strongly affects the network’s reconstruction ability.

In our study, we used synthetic data to prove the feasibility of CCEIT for human brain imaging. The obtained results allow us to conclude that it is possible to use the presented approach to generate synthetic datasets and successfully train ANN for image reconstruction using such datasets. However, the obtained results should be further verified by reconstructing images from real measurements of physical phantoms before it could advance toward the adoption of this technique into clinical practice. In our future research, we plan to enhance our model with a classifier convolutional network to evaluate the possibility of detecting specific lesions using reconstructed images.

However, we are continuing our research on possible network architecture enhancements to mitigate measurement disruptions introduced by the skull. Among such enhancements, we consider fine-tuning training parameters such as learning rates and their adjustments during the training procedure. Another option could be switching to a complex-domain-based network because it deals with electrical properties represented by complex numbers. We have shown how using the real and imaginary parts of measurements as two real numbers is a working approach. Still, we suspect that processing both components as a complex number can improve the reconstruction algorithm’s robustness.

## 7. Conclusions

Based on numerical simulation, we can conclude that CCEIT measurements combined with neural network image reconstruction have the potential for conductivity spatial distribution imaging in the human head. The differences in dielectric properties reflected in the proposed test object correspond to the changes observed in the brains of stroke-affected patients. Due to conductivity differences occurring in different types of strokes (hemorrhagic and ischemic), it may be possible to distinguish between them based on CCEIT measurement results. The image reconstructions we achieved allowed us to deduce that this method enables the effective imaging of such minor changes. In our opinion, it makes sense to explore the potential of CCEIT as a new diagnostic device.

## Figures and Tables

**Figure 1 life-14-00419-f001:**
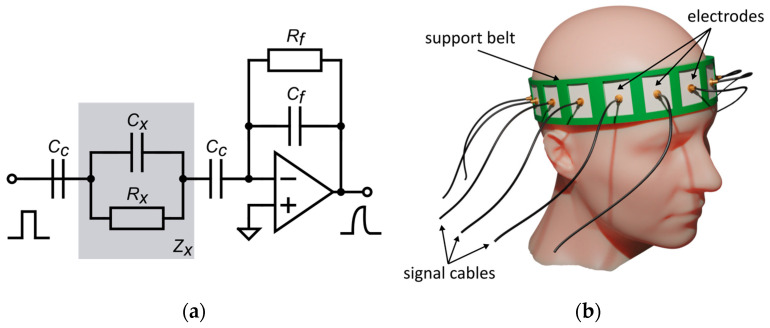
(**a**) The CCEIT measurement system with pulse excitation; (**b**) the measuring belt on the patient’s head.

**Figure 2 life-14-00419-f002:**
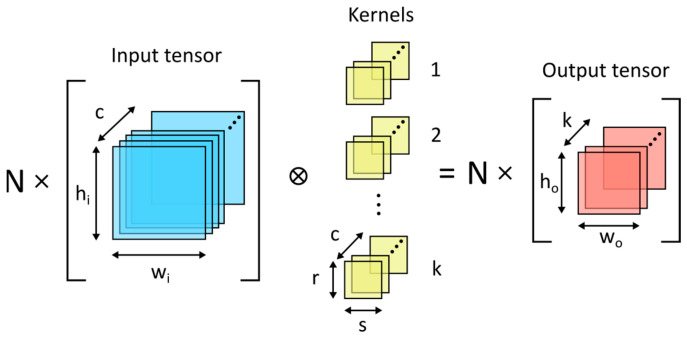
Two-dimensional convolution layer: N—batch size; c—number of input channels; h_i_, w_i_—input matrix size; r, s—kernel size; h_o_, w_o_—output matrix size; k—number of kernels; and ⊗—convolution operation.

**Figure 3 life-14-00419-f003:**
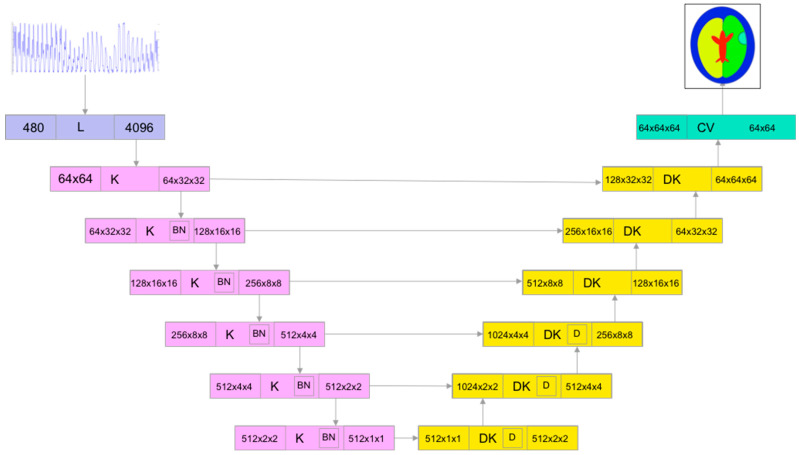
Generator network architecture for brain image reconstruction: K—convolutional block, DK—deconvolutional block, CV—convolutional layer, and L—linear layer [15].

**Figure 4 life-14-00419-f004:**
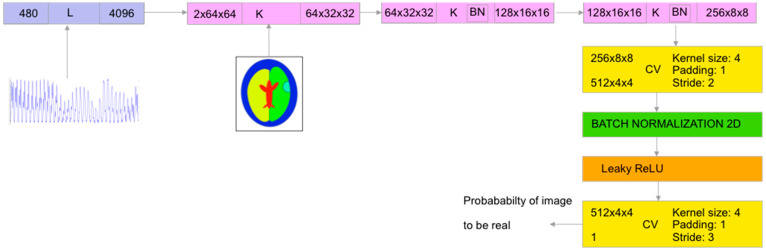
Discriminator network architecture for brain image reconstruction: K—convolutional block, DK—deconvolutional block, CV—convolutional layer, and L—linear layer [15].

**Figure 5 life-14-00419-f005:**
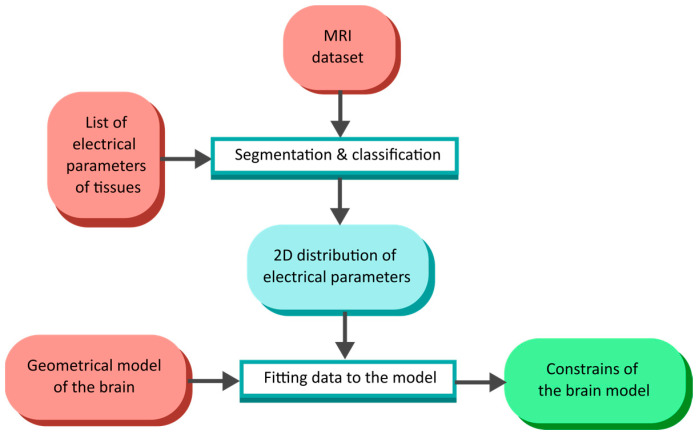
Flow diagram of numerical phantom creation based on the MRI examination dataset.

**Figure 6 life-14-00419-f006:**
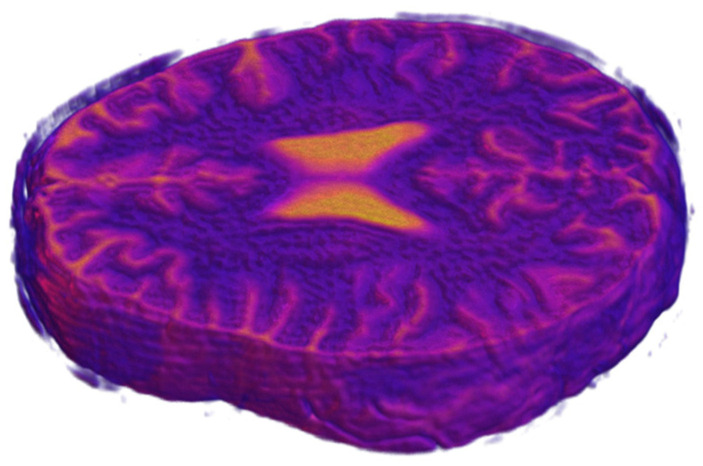
Example of three centimeter cross-section of a human brain cut out from T1 MRI 3D image.

**Figure 7 life-14-00419-f007:**
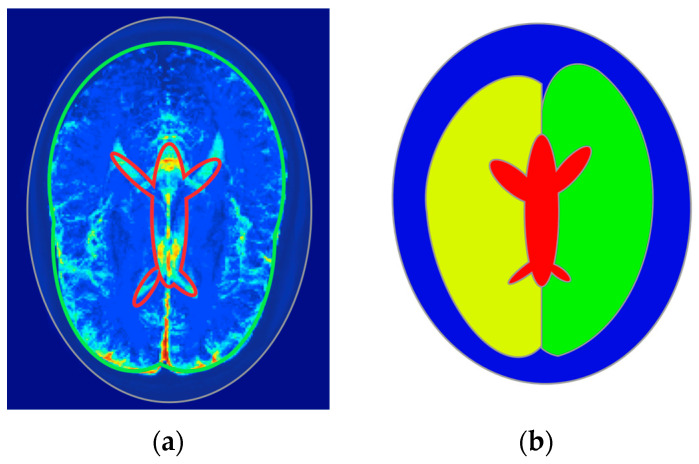
(**a**) Mean conductivity cross-section of a healthy person’s head with marked regions. (**b**) The geometry of the numerical phantom of the head of a healthy person.

**Figure 8 life-14-00419-f008:**
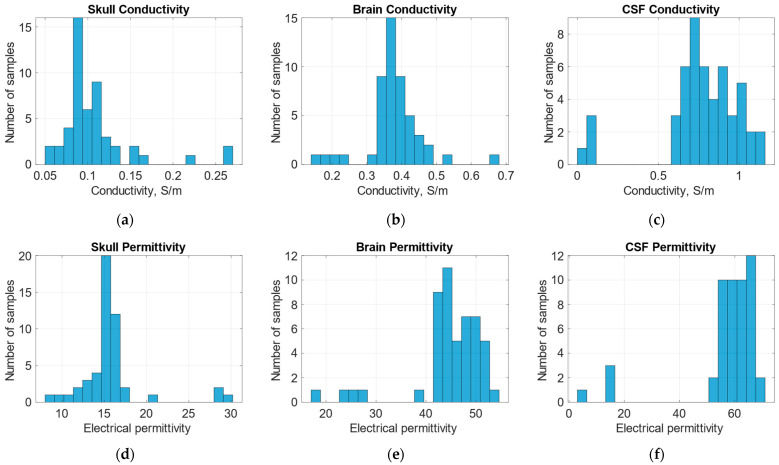
Conductivity and permittivity distribution throughout 50 selected samples in regions of interest such as the skull, brain hemispheres, and cerebrospinal fluid (CSF): (**a**) distribution of the conductivity in the skull region; (**b**) distribution of the conductivity in the brain region; (**c**) distribution of the conductivity in the CSF region; (**d**) distribution of the electrical permittivity in the skull region; (**e**) distribution of the electrical permittivity in the brain region; and (**f**) distribution of the electrical permittivity in the CSF region.

**Figure 9 life-14-00419-f009:**
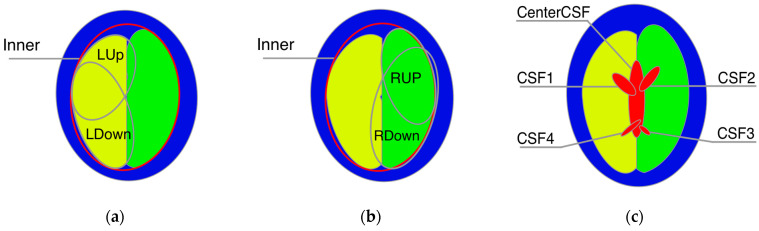
Geometry elements of the numerical phantom of the head: (**a**) the left hemisphere is the intersection of the ‘Inner’ object with the union of ‘LUp’ and ‘LDown’ objects, (**b**) the right hemisphere is the intersection of the ‘Inner’ object with the union of ‘Rup’ and ‘RDown’ objects, and (**c**) cerebrospinal fluid is the union of ‘CenterCSF’, ‘CSF1–CSF4’ objects.

**Figure 10 life-14-00419-f010:**
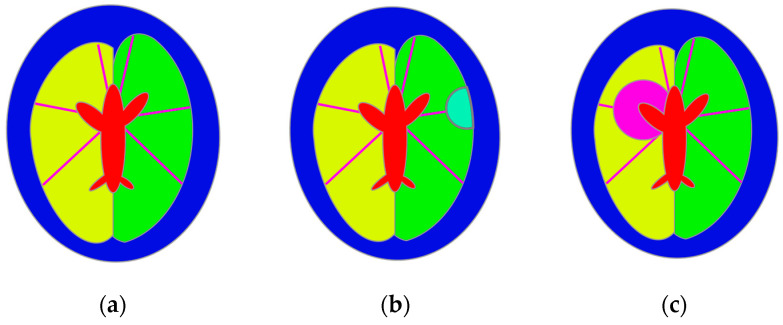
Model of the head of the patient affected by stroke: (**a**) main brain arteries—the possible center of the stroke, (**b**) ischemic stroke, and (**c**) hemorrhagic stroke.

**Figure 11 life-14-00419-f011:**
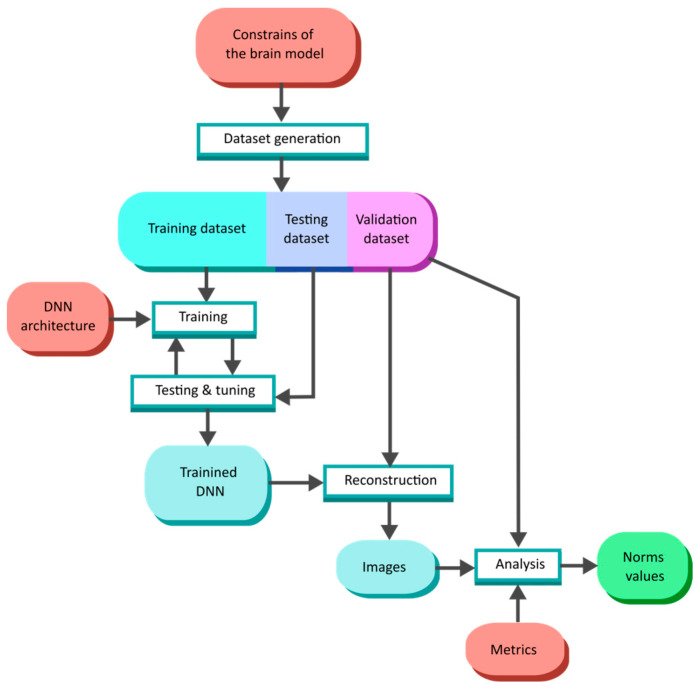
Flow chart of neural network training, testing, and validation.

**Figure 12 life-14-00419-f012:**
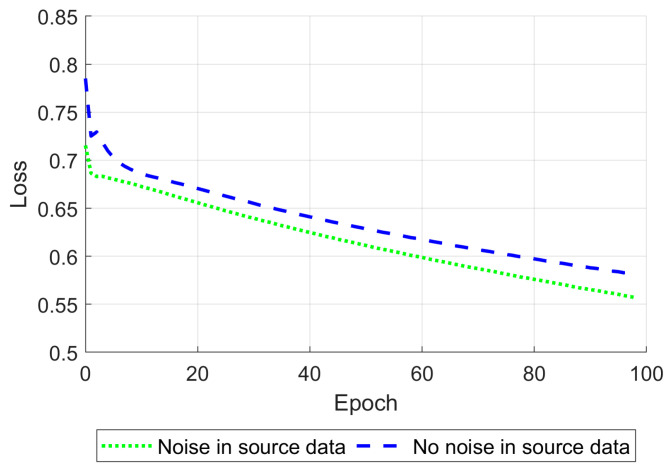
Discriminator loss function on 100 epochs for generator and discriminator starting learning rates set to 10^−3^ and 10^−8^, respectively, shown for the training on data without (blue) and with (green) noise.

**Figure 13 life-14-00419-f013:**
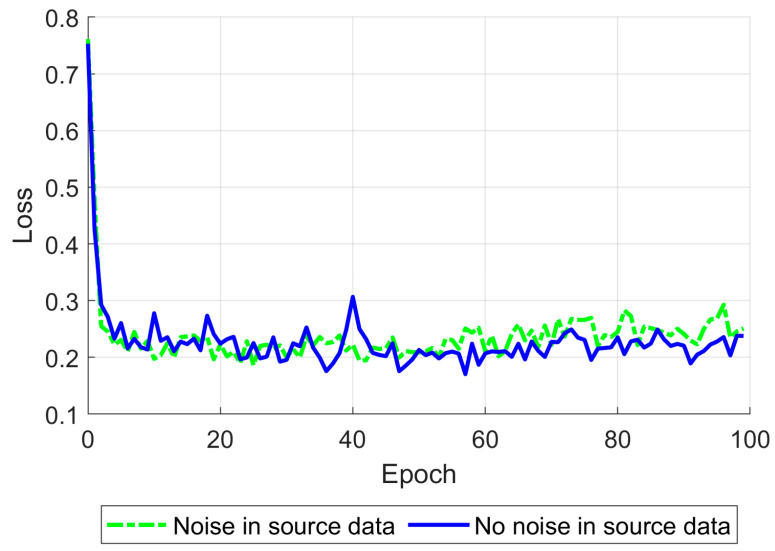
The mean relative standard error for 100 epochs is shown for the training on data without (blue) and with (green) noise.

**Figure 14 life-14-00419-f014:**
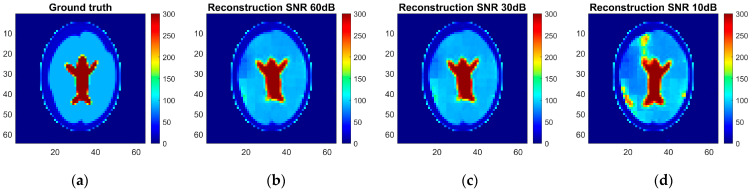
Ground truth and reconstructed conductivity values for the healthy patient model for various noise levels added to the measurement data: (**a**) ground truth, (**b**) measurement data with 60 dB SNR, (**c**) measurement data with 30 dB SNR, and (**d**) measurement data with 10 dB SNR.

**Figure 15 life-14-00419-f015:**
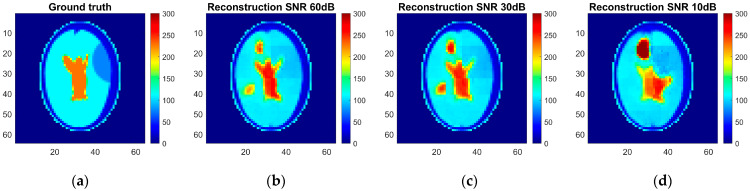
Ground truth and reconstructed conductivity values for the ischemic stroke-affected patient model for various noise levels added to the measurement data: (**a**) ground truth, (**b**) measurement data with 60 dB SNR, (**c**) measurement data with 30 dB SNR, and (**d**) measurement data with 10 dB SNR.

**Figure 16 life-14-00419-f016:**
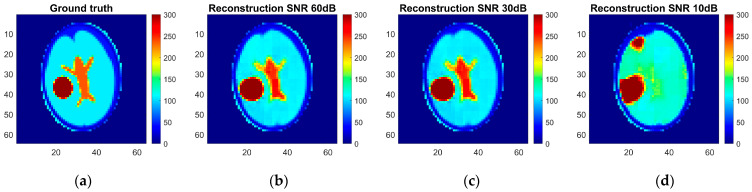
Ground truth and reconstructed conductivity values for the hemorrhagic stroke-affected patient model for various noise levels added to the measurement data: (**a**) ground truth, (**b**) measurement data with 60 dB SNR, (**c**) measurement data with 30 dB SNR, and (**d**) measurement data with 10 dB SNR.

**Figure 17 life-14-00419-f017:**
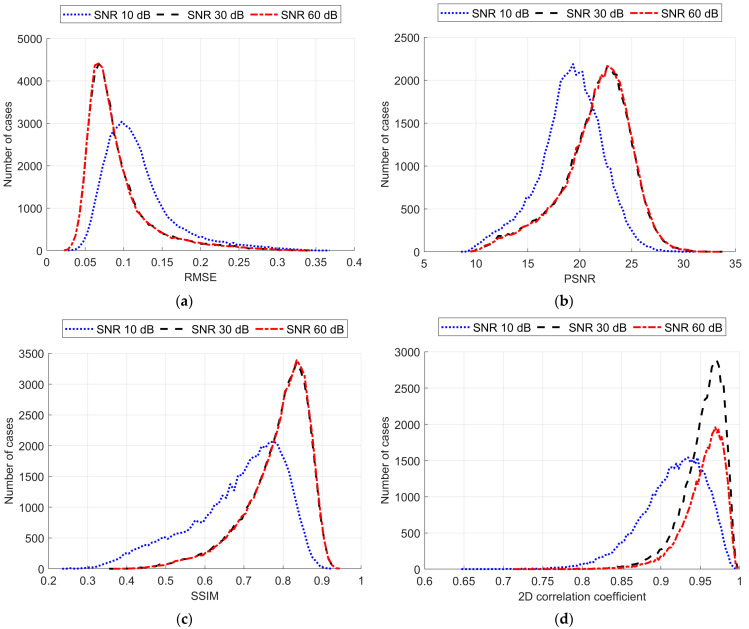
Reconstruction metrics distribution for various noise levels added to the measurement data as follows: (**a**) root mean square error, (**b**) peak noise-to-signal ratio, (**c**) structural similarity index (SSIM), and (**d**) 2D correlation coefficient.

**Table 1 life-14-00419-t001:** Dielectric properties of brain tissues at 64 MHz. Permittivity and conductivity values are taken form [55].

Tissue	ε_r_	$\sigma \left[\frac{mS}{m}\right]$
White matter	67.8	292
Grey matter	97.4	511
Bones	16.7	59.5
Cerebrospinal fluid	97.3	2070
Other (fat)	13.6	66.2
Skin	92.2	436

**Table 2 life-14-00419-t002:** Calculated dielectric properties of model regions at 64 MHz.

Region	ε_r_		$\sigma \;\left[\frac{mS}{m}\right]$	
	Mean	Std	Mean	Std
Skull and skin	14.79	1.77	102	29
Brain	46.46	3.44	378	59
Cerebrospinal fluid	61.26	4.40	788	207

**Table 3 life-14-00419-t003:** Geometric constraints for elements of the brain numerical phantom.

Element	Object	Relative to Object	Center Position [mm]		Size [mm]		Tilt [deg]
			x	y	x	y	
Skull	Head		2.0 ± 3.0	14.0 ± 8.0	91.0 ± 20.0	99.0 ± 12.0	−2.5 ± 8.5
Brain	Inner	Head	−0.5 ± 2.5	−1.5 ± 4.5	−14.0 ± 5.0	−14.0 ± 5.0	2.0 ± 10.0
	LDown	Inner	−25.5 ± 9.5	−21.5 ± 9.5	−23.5 ± −14.5	−50.0 ± 12.0	−72.5 ± 12.5
	LUp	Inner	−24.5 ± 7.5	24.0 ± 10.0	−30.5 ± 7.5	−32.0 ± 9.0	−20.0 ± 22.0
	LUp	LDown	0.0 ± 8.0	49.0 ± 13.0	−4.5 ± 37.5	21.0 ± 12.0	48.5 ± 17.5
	RDown	Inner	25.5 ± 6.5	−20.0 ± 11.0	−25.5 ± −17.5	−54.5 ± 11.5	72.5 ± 11.5
	RDown	LDown	47.0 ± 6.0	0.0 ± 3.0	1.0 ± 5.0	0.5 ± 5.5	151.5 ± 13.5
	RUp	Inner	27.0 ± 7.0	26.0 ± 13.0	−32.0 ± 9.0	−33.5 ± 7.5	29.0 ± 15.0
	RUp	RDown	3.0 ± 9.0	48.0 ± 16.0	−31.5 ± 14.5	22.0 ± 8.0	−46.0 ± 17.0
	RUp	LDown	52.5 ± 12.5	0.0 ± 3.0	0.0 ± 11.0	−3.0 ± 8.0	55.0 ± 23.0
CSF	Center CSF	Inner	0.5 ± 2.5	−2.5 ± 13.5	−54.5 ± 8.5	−53.0 ± 15.0	2.0 ± 9.0
			**Axis angle [deg]**	**Shift** **[mm]**	**Size [mm]**		**Tilt** **[deg]**
					**x**	**y**	
	CSF1		149.5 ± 10.5	8.5 ± 1.5	17.5 ± 2.5	9.0 ± 1.0	−27.0 ± 6.0
	CSF2		25.0 ± 23.0	8.5 ± 1.5	18.0 ± 2.0	9.0 ± 1.0	22.0 ± 5.0
	CSF3		−51.0 ± 11.0	1.5 ± 4.5	13.0 ± 5.0	5.5 ± 2.5	5.5 ± 5.5
	CSF4		−130.5 ± 13.5	1.5 ± 4.5	16.0 ± 5.0	5.0 ± 2.0	−6.0 ± 6.0

**Table 4 life-14-00419-t004:** Circumference and axes ratio constraints for elements of the brain numerical phantom.

Element	Object	Circumference [mm]	Axes Ratio
Skull	Head	570.0 ± 70.0	1.2 ± 0.2
Brain	Inner	487.5 ± 53.5	1.3 ± 0.2
	LDown	0.7 ± 0.1 *	0.6 ± 0.1
	LUp	1.3 ± 0.1 **	1.8 ± 0.5
	RDown	0.6 ± 0.1 *	0.5 ± 0.1
	RUp	1.3 ± 0.1 ***	1.6 ± 0.5
CSF	CenterCSF	0.3 ± 0.1 *	4.4 ± 2.9
	CSF1	93.5 ± 24.5	0.3 ± 0.2
	CSF2	94.0 ± 22.0	0.4 ± 0.2
	CSF3	57.0 ± 21.0	0.5 ± 0.3
	CSF4	70.0 ± 23.0	0.3 ± 0.3

*—relative to the inner object circumference, **—the sum of LDown and LUp circumferences relative to the inner object circumference, ***—the sum of RDown and Rup circumferences relative to the inner object circumference.

**Table 5 life-14-00419-t005:** Image quality metrics for normalized images calculated on a verification set depending on SNR.

Measure	SNR 60 dB		SNR 30 dB		SNR 10 dB	
	Mean	Std	Mean	Std	Mean	Std
RMSE	0.09	0.04	0.09	0.04	0.12	0.05
PSNR	21.58	3.39	21.53	3.39	19.08	3.08
SSIM	0.81	0.08	0.81	0.08	0.71	0.12
2D correlation	0.95	0.03	0.95	0.03	0.92	0.04

## Data Availability

Data are available on request.

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
