# Peer review of "Generative-Adversarial-Network-Based Image Reconstruction for the Capacitively Coupled Electrical Impedance Tomography of Stroke"

_life, 2024, doi:10.3390/life14030419_

Round 1

Reviewer 1 Report

Comments and Suggestions for Authors

The authors demonstrated the reconstruction of CCEIT on numerically simulated brains by cGAN algorithm. The whole study is powered by simulation because this is a pioneer study. The flow of the article is adequate and needs some minor revision.

1. A flow diagram for the whole study design, including the generation of phantoms and training/validation, would make the article more readable.

2. In lines 88-97(2.1), the reason for using numerical simulation could be moved to the "discussion" section, making "material and methods" more compact.

3. Figures 4,11,12,13, the tiny figures should be labeled according to the format of Life.

4. Did you intend to quantitatively evaluate the ability to detect lesions with CCEIT with cGAN reconstruction in this article?

5. How did you determine the quantity of simulated cases? Did you calculate the alpha?

6. Are there other algorithms for CCEIT/EIT reconstruction? If so, please compare or state in the discussion.

7. In the "discussion" section, please discuss the strengths and limitations of your work point-to-point.

Comments on the Quality of English Language

The wording could be more compact.

Author Response

  1. A flow diagram for the whole study design, including the generation of phantoms and training/validation, would make the article more readable

Indeed, our article lacked a clear graphical representation of the flow chart, which could have improved readability. Following your suggestion, we have addressed this issue by dividing the description of our method's implementation into two stages. We have prepared a new UML diagram for each stage that sequentially illustrates our process. We believe these additions will significantly enhance the reader's understanding of our study design, including the generation of phantoms and the training/validation process.

  1. In lines 88-97(2.1), the reason for using numerical simulation could be moved to the "discussion" section, making "material and methods" more compact

Thank you for your suggestion. We recognize that the Materials and Methods section was unnecessarily extensive upon review. Following your recommendation, we have decided to restructure the article by excluding the "Numerical Phantom" and "Network training" sections from the "Materials and Methods Section." We believe that this should improve article clarity and coherence.

  1. Figures 4,11,12,13, the tiny figures should be labeled according to the format of Life

Thank you for pointing this out. We fixed that error.

  1. Did you intend to quantitatively evaluate the ability to detect lesions with CCEIT with cGAN reconstruction in this article?

The article is focused mainly on the applicability of the cGAN approach to image reconstruction in brain CCIET. Your suggestion for evaluating the ability to detect lesions using reconstructed images is very interesting, and we will include this in our plan for further studies. We added that information to the Discussion section.

  1. How did you determine the quantity of simulated cases? Did you calculate the alpha?

Calculating the required dataset size for neural network training is possible using the Vapnik–Chervonenkis dimension in the case of fully connected neural networks. Still, unfortunately, when we talk about more complex networks like cGAN, the task becomes much more complicated. In our study, we determined the quantity of simulated cases through an empirical approach. However, we are actively working on defining the optimal size of the training dataset.

  1. Are there other algorithms for CCEIT/EIT reconstruction? If so, please compare or state in the discussion

References to foundational reconstruction algorithms are included in the Introduction. We have not provided an extensive description of these methods as we do not perform a direct comparison within our study. Nevertheless, we have added a reference to algebraic methods in the "Discussion" section, offering a brief comparative perspective. This addition aims to acknowledge the existence of other algorithms and provide readers with a context for understanding how our approach relates to or differs from these methods.

  1. In the "discussion" section, please discuss the strengths and limitations of your work point-to-point

Following your suggestion, we have incorporated a new paragraph addressing our study's strengths and limitations in the "Discussion" section. Additionally, we have outlined potential directions for future research.

Reviewer 2 Report

Comments and Suggestions for Authors

The work is interesting

Suggestions:

it would be useful for the readers to be included a section about the machine learning and more informations about the methods that the authors are using including equations and graphs

Author Response

  1. it would be useful for the readers to be included a section about the machine learning and more informations about the methods that the authors are using including equations and graphs

Thank you for your comment, which highlighted a potential issue with the clarity of our Methods section. To facilitate a better understanding for the reader, we have restructured the article by dividing the Methods section into three distinct subsections. This revision has allowed us to provide a more detailed description of our method. Additionally, we have included three new figures that offer a more transparent graphical representation of our study workflow. We believe these changes significantly improve the manuscript's readability and allow for a more comprehensive grasp of the methodologies employed.

Reviewer 3 Report

Comments and Suggestions for Authors

This paper deals with a new method of image reconstruction in capacitively coupled tomography. The authors generated a synthetic dataset of measurements and related images and trained the neural reconstruction system. The reconstructed images turn out to be very close to the ground truth used in the training phase, highlighting the possible potential of the proposed approach.

This Reviewer appreciates the authors' effort but thinks an effort is needed to strengthen this contribution. Below are some comments that may improve the presentation of the paper:

- the abstract should contain a spoiler on the main results regarding the performance achieved.

- The state of the art deserves more in-depth discussion. The authors only report that reconstructing images with neural networks or other AI methodologies is possible but do not mention the advantages and limitations of the cited work. The rationale behind the need for this study should be better expressed based on the limitations and needs that emerged from the literature study.

- Section 2 should include an introduction in which the section's content is presented. How is the proposed processing pipeline implemented? How is validation carried out? In this way, the reader is clear about the authors' process before going into the details of the individual components.

- is there no way to apply the proposed methodology to real data? This I think is my biggest criticism of the authors. If this is not possible, the authors should discuss the limitations of what they are presenting in an appropriate section and how they expect performance to change when moving to an operational environment with real, not synthetic, data.

- Table 5 can directly have the column with the metrics reported as mean \pm std

- Figure 14 should have in each graph the xlabel of the metric under consideration (with its unit of measurement) and not in the title

Author Response

  1. the abstract should contain a spoiler on the main results regarding the performance achieved.

Thank you for pointing out the need to include a summary of the main results and performance achievements in the abstract. In response, we have added sentences to the abstract that precisely outline our study's key findings and conclusions.

  1. the state of the art deserves more in-depth discussion. The authors only report that reconstructing images with neural networks or other AI methodologies is possible but do not mention the advantages and limitations of the cited work. The rationale behind the need for this study should be better expressed based on the limitations and needs that emerged from the literature study

Thank you for highlighting this issue. To address your concerns, we have revised the paragraph regarding our closest competitors in the Introduction to more clearly articulate what novel contributions our article brings to the state of the art. Additionally, in the Discussion section, we have added a paragraph that positions our method within the context of others, considering its capabilities and limitations. This enhancement aims to provide a more comprehensive understanding of the rationale behind our study, grounded in the limitations and needs identified through our literature review.

  1. Section 2 should include an introduction in which the section's content is presented. How is the proposed processing pipeline implemented? How is validation carried out? In this way, the reader is clear about the authors' process before going into the details of the individual components.

Thank you for your comment, which has made us realize that the Methods section might have been inadequately structured for optimal understanding. To address this, we have reorganized the article's structure, dividing the Methods section into three more concise subsections. This reorganization allows for a more detailed description of our method. Additionally, we have included three new figures to visually depict the flow chart of our work more transparently. We believe these modifications significantly enhance the article's readability and provide readers with a more precise overview of the process before delving into the specifics of the individual components.

  1. Is there no way to apply the proposed methodology to real data? This I think is my biggest criticism of the authors. If this is not possible, the authors should discuss the limitations of what they are presenting in an appropriate section and how they expect performance to change when moving to an operational environment with real, not synthetic, data.

Thank you for pointing out a potential misunderstanding that may have arisen from an unclearly written section of our paper. Indeed, applying our method to real data is possible, and we intend to conduct experiments with real data in future research.

Using numerical simulation data is crucial for preparing a training dataset that requires a much larger number of cases than could feasibly be obtained from patient studies. However, thanks to our CCEIT measurement simulation system, ectSIM, we can achieve simulations with high accuracy that replicate actual measurements obtained using our tomographic system.

In this paper, we validate the reconstruction results of our network using numerical data. However, in future work, we plan to design and conduct tests using ballistic gel phantoms that simulate the electrical conductivity distribution of the human brain. If these studies prove successful, we will endeavor to obtain reconstructions based on measurements from actual patients.

We have tried to revise the article's sections that were unclear and have incorporated the information presented in this response into our manuscript.

  1. Table 5 can directly have the column with the metrics reported as mean \pm std

Thank you for your suggestion regarding the presentation of metrics in Table 5 using the mean ± standard deviation format. We understand the rationale behind your recommendation, as this notation is commonly employed in scientific reporting. However, we have chosen to adhere to our original presentation format. This decision is based on feedback and considerations that the mean ± standard deviation notation, despite its prevalence, can sometimes lead to misunderstandings or misinterpretations. Interestingly, this topic is discussed in an article on the website https://www.ncbi.nlm.nih.gov/pmc/articles/PMC2959222/, which highlights the potential issues and considerations associated with this notation.

  1. Figure 14 should have in each graph the xlabel of the metric under consideration (with its unit of measurement) and not in the title

Thank you for your observation regarding the labeling of Figure 14. Following your advice, we have revised the figure's captions to ensure that each graph now includes the xlabel and the ylabel of the metric under consideration, along with its unit of measurement, rather than incorporating this information in the title. This adjustment aligns with the stylistic guidelines of the journal.

Round 2

Reviewer 3 Report

Comments and Suggestions for Authors

No other comments.